# Scalable Multi-Modal Continual Meta-Learning

## Abstract

This paper focuses on continual meta-learning, where few-shot tasks from a non-stationary distribution are sequentially available. Recent works maintain a mixture distribution of meta-knowledge to cope with the heterogeneity and a dynamically changing number of components in the mixture distribution to capture incremental information. However, the underlying assumption of mutual exclusiveness among mixture components hinders sharing meta-knowledge across different tasks. Another issue is that they only use a prior to determine whether to increase meta-knowledge components, leading to parameter inefficiency. In this paper, we propose a Scalable Multi-Modal Continual Meta-Learning (SMM-CML) algorithm, which employs a multi-modal premise to encourage different clusters of tasks to share meta-knowledge. Specifically, every task cluster is associated with a subset of mixture components, which is achieved by an Indian Buffet Process prior. Besides, to avoid parameter inefficiency caused by the unlimited increase, we propose a component sparsity method based on evidential theory to learn the posterior number of components, filtering out those meta-knowledge without receiving support directly from tasks. Experiments show SMM-CML outperforms strong baselines, which illustrates the effectiveness of our multi-modal meta-knowledge, and confirms that our algorithm can learn parameter-efficient meta-knowledge.

## 1 Introduction

Meta-learning (Vanschoren, 2018; Hospedales et al., 2020) is widely used in the low-resource setting. The key idea is to transfer meta-knowledge (i.e., the experience about how to learn) to improve data efficiency and enhance model generalization. In contrast to the conventional assumption that data are homogeneous and available at once (Finn et al., 2017; 2018), continual meta-learning faces a more practical setting where data are heterogeneous and sequentially available (Finn et al., 2019; Denevi et al., 2019). That is, tasks from non-stationary distributions arrive sequentially. There are two challenges to consider in this setting: (1) to avoid forgetting the learned meta-knowledge when training on tasks sampled from the heterogeneous distribution, also called as *catastrophic forgetting* (Kirkpatrick et al., 2017); (2) to capture the incremental meta-knowledge when encountering the newer tasks (Lee et al., 2017).

For the first challenge, existing works (Jerfel et al., 2019; Yao et al., 2019; Zhang et al., 2021) use a mixture model, associating a cluster of similar tasks with a single component. One major concern is that they implicitly assume different meta-knowledge components are mutually exclusive. This assumption impedes the sharing of meta-knowledge among clusters of tasks, which could lead to suboptimal performance and bias toward one type of meta-knowledge. For example, in the research of user profiling, a user (i.e., a task) can belong to multiple preference groups (i.e., components), so if modeling by a single meta-knowledge component, the algorithm might focus more on one preference and result in biased profiling.

For the second challenge, these works (Jerfel et al., 2019; Yao et al., 2020; Zhang et al., 2021) incrementally update meta-knowledge, where a new meta-knowledge component is added to the mixture model for new tasks. However, all of them just leverage a prior (Jerfel et al., 2019; Zhang et al., 2021) or make a simple judgment before the update of meta-knowledge (Yao et al., 2019; 2020) on whether to add new meta-knowledge components but cannot make a posterior decision from task

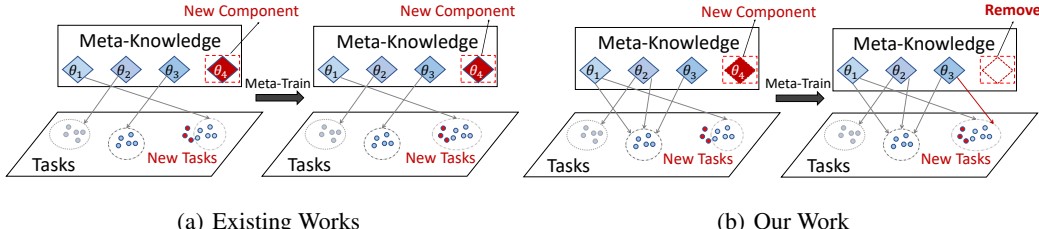

(a) Existing Works  (b) Our Work

Figure 1: The difference in incremental meta-knowledge between the existing works and ours. Previous methods only make a prior decision on whether to add new meta-knowledge (the red dashed grid), which might reproduce the redundant components. Our algorithm considers making a posterior decision from tasks after the meta-train to filter out the meta-knowledge without receiving support.

data. Also, they can only increase but is not able to decrease the number of mixture components as needed, leading to parameter inefficiency when meeting a large number of tasks (seen Fig. 1).

To solve these problems, we propose a Scalable Multi-Modal Continual Meta-Learning algorithm, abbreviated as SMM-CML. The proposed SMM-CML associates a task cluster with a subset of components of the meta-knowledge mixture model, where the provided meta-knowledge is multi-modal (i.e., a statistical distribution of values with multiple peaks) with each mode being a related meta-knowledge component. The multi-modal meta-knowledge relaxes the constraint of a single component, so that it allows different clusters of tasks to share the meta-knowledge via the overlapped components. This is achieved by employing the Indian Buffet Process (IBP) prior on the number of components when meeting new tasks. To correct the prior after the update of meta-knowledge on new tasks, we propose an evidential sparsification method to decide the posterior number of components, filtering out the meta-knowledge which does not receive support information directly from task data. Our contributions are summarized as:

- We propose multi-modal meta-knowledge, where a task is associated with a subset of components of meta-knowledge mixture model instead of a single one. Our multi-modal premise allows sharing meta-knowledge via the overlapped components among different clusters of tasks so as to avoid bias towards one type of meta-knowledge.

- We employ the IBP prior to allow the number of mixture components to increase with the newer task arriving, and propose an evidential sparsification method to learn the posterior number of components from tasks, filtering out the meta-knowledge which does not receive support information directly from all occurring tasks. The combination of IBP and evidential sparsification helps to maintain the scalable meta-knowledge to cope with the online non-stationary setting.

- We conduct extensive experiments and the results show that our SMM-CML outperforms the-state-of-art baselines under the online non-stationary setting. And it also confirms the effectiveness of multi-modal meta-knowledge and that our algorithm can learn the parameter-efficient meta-knowledge from tasks.

## 2 RELATED WORK

**Meta-Learning.** Meta-learning (Vanschoren, 2018; Hospedales et al., 2020) focuses on a few-shot setting. It assumes that source tasks can be used to help with the learning in the target tasks. Recent works include metric-based(Snell et al., 2017; Oreshkin et al., 2018), model-based(Ha et al., 2016; Munkhdalai & Yu, 2017), optimization-based methods(Finn et al., 2017; 2018) and their Bayesian variants (Ravi & Beatson, 2018; Gordon et al., 2019; Iakovleva et al., 2020), respectively. However, most of them propose to construct a globally-shared meta-knowledge, which can not fit the heterogeneous data distribution in the real world (Jerfel et al., 2019). To solve this problem, some works (Jerfel et al., 2019; Zhang et al., 2021) maintain a mixture of meta-knowledge, where a cluster of similar tasks is associated with a single component of the meta-knowledge. This impedes the sharing of meta-knowledge between different clusters of tasks. Different from the existing works, we take into account both sharing and diversity of meta-knowledge simultaneously.

**Continual Learning.** Conventional continual learning (Delange et al., 2021) concentrates on the large-scale data setting. Existing models prevent the catastrophic forgetting issue via replay (Hu et al.,

2019; Titsias et al., 2019), regularization (Benjamin et al., 2018; Pan et al., 2020) and incremental model selection (Kumar et al., 2021; Kessler et al., 2021). Recently, many works based on meta-learning (Finn et al., 2019; Zhuang et al., 2020) focus on the low-resource setting. Inspired by the incremental model selection, some existing works extend meta-knowledge when encountering new tasks, via increasing the number of mixture components (Yao et al., 2019) or adding a novel block to construct the mate-path (Yao et al., 2020). Moreover, the Chinese Restaurant Process (CRP) has been used to determine the prior number of meta-knowledge components (Jerfel et al., 2019; Zhang et al., 2021). However, these methods only consider how to construct the prior number of components and do not make a posterior decision from tasks. Such a prior determination only allows the increase of meta-knowledge, which would lead to parameter inefficiency and large computational consumption. In our work, we learn the posterior number of meta-knowledge components from tasks via the combination of IBP prior and the evidential sparsification method.

**Sparsification Method**   In recent years, a number of methods have been proposed to sparse the multi-modal space. Most of them (Martins & Astudillo, 2016; Laha et al., 2018) aim to propose a softmax alternative to sparse the large output space. Itkina et al. (2020) pointed out that the above methods are aggressive, and propose a post hoc evidential sparsification for conditional variational auto-encoder, based on the conclusion in (Denœux, 2019) that most existing classifiers can be seen as converting features into mass function and merging them to the final result. Following Itkina et al. (2020), Chen et al. (2021) presented evidential softmax method. However, these methods operate on mutual exclusiveness, which is in conflict with ours. Moreover, our evidential sparsification method provides a novel view of how to apply the evidential theory on continual learning.

## 3 BACKGROUND

### 3.1 BAYESIAN ONLINE META-LEARNING

Suppose there are sequentially arriving tasks $\tau_t$ with a dataset $\mathcal{D}_t$ from a non-stationary distribution $p(\tau)$. Note that the dataset $\mathcal{D}_t$ is split into two sub-datasets, a support set $\mathcal{D}_t^S = \{x_i, y_i\}_{i=1}^{N_t}$ for training and a query set $\mathcal{D}_t^Q = \{x_i, y_i\}_{i=1}^{M_t}$ for validation.

Catastrophic forgetting (Lee et al., 2017) is a key issue in continual learning. To overcome such an issue in the non-stationary task flow, some variational methods (Yap et al., 2021; Zhang et al., 2021) have been developed. They update the meta-knowledge in an online way following Variational Continual Learning (VCL)(Nguyen et al., 2018):

$$p(\theta_t|\mathcal{D}_{1:t}) \propto p(\mathcal{D}_t|\theta_t)p(\theta_t|\mathcal{D}_{1:t-1}), \tag{1}$$

where $\theta_t$ is the meta-knowledge and used as the initialization following MAML (Finn et al., 2017). Note that it assumes that the datasets $\mathcal{D}_{1:t}$ are independent given $\theta_t$. Thus, the meta-knowledge can be updated in a recursively way. Then, the meta-learning framework can be reformulated as a variational way (Gordon et al., 2019; Iakovleva et al., 2020):

$$p(\mathcal{D}_t|\theta_t) = \int p(\mathcal{D}_t|\phi_t)p(\phi_t|\theta_t)d\phi_t, \tag{2}$$

where $\phi_t$ is the task-specific parameter. To learn such intractable posteriors, some inference methods (e.g., variational inference (Kingma & Welling, 2013)) are applied to infer the approximate distributions. More details of inference are in Appendix A.

### 3.2 EVIDENTIAL THEORY

Evidential theory (Denœux, 2019) works on a discrete set of hypotheses (or equivalently, components of meta-knowledge in this paper). Let $Z = \{z_1, z_2, z_3, ..., z_K\}$ be a finite set, the element of which $z_k$ is a binary variable indicating whether the current task is associated with $k$-th component or not, and the power set of $Z$, donated by $2^Z$. A *mass function* on $Z$ is a mapping $m: 2^Z \rightarrow [0, 1]$ and satisfies the following constraints:

$$m(\emptyset) = 0, \quad \sum_{A \subseteq Z} m(A) = 1. \tag{3}$$

Figure 2: The framework of SMM-CML. The top is the updating of meta-knowledge, where the IBP prior determines whether to add new meta-knowledge components (the red box), and the evidential sparsity method is used to filter out the component without receiving support directly from tasks based on the posterior of beta distribution from the current time (the solid line) and the previous time (the dashed line). The bottom is the task-specific adaption, where the multi-modal meta-knowledge is decided and then adapted to the task-specific parameter based on the support set $\mathcal{D}_t^S$ (the dashed line), and then the task-specific parameter is used to make the prediction on the data (the solid line).

The mass function $m(\cdot)$ represents the support to each potential subset of components provided by a piece of evidence, and any subset $A$ is called *focal set* if $m(A) > 0$. As a particular case, the vacuous mass function (i.e., $m(Z) = 1$) indicates that the evidence can not provide any information. One mass function is said *simple* when:

$$m(A) = s, \quad m(Z) = 1 - s, \quad w = -\ln(1 - s), \tag{4}$$

where $A$ is a single strict subset $A \subset Z$, $s \in [0, 1]$ represents the support degree of $A$, and $w$ donates the *evidential weight* of $A$.

Given a mass function, there are two corresponding functions, called *belief and plausibility function*, respectively, which are defined as follows:

$$Bel(A) = \sum_{B \subseteq A} m(B), \quad Pl(A) = \sum_{B \cap A \neq \emptyset} m(B) = 1 - Bel(\bar{A}). \tag{5}$$

$Bel(A)$ can be interpreted as the total support degree to $A$, while the $1 - Pl(A)$ can be interpreted as the total doubt degree to $A$. Besides, when the plausibility function is restricted to singletons, then it is called *contour function* $pl : z_k \rightarrow [0, 1]$.

Given two mass functions provided by different evidence, the fusion of them follows Dempster's rule (Dempster, 2008). More details about the computing rules are in Appendix B.

## 4  SCALABLE MULTI-MODAL CONTINUAL META-LEARNING

In this section, we present our Scalable Multi-Modal Continual Meta-Learning algorithm (SMM-CML). The total framework of SMM-CML is seen in Fig. 2.

### 4.1  MULTI-MODAL CONTINUAL META-LEARNING

SMM-CML relaxes the constraint that a task is associated with only a single component of meta-knowledge. The restriction of one-to-one mapping prevents the sharing of meta-knowledge among

different clusters of tasks. We employ the multi-modal meta-knowledge where multiple meta-knowledge components are maintained. It assumes that a cluster of similar tasks is associated with a subset of components of meta-knowledge:

$$p(\mathcal{D}_t|\theta_t) = \int p(\mathcal{D}_t|\theta_t, \boldsymbol{z_t})p(\boldsymbol{z_t})d\boldsymbol{z_t} = \int \left[ \int p(\mathcal{D}_t|\phi_t)p(\phi_t|\theta_t, \boldsymbol{z_t})d\phi_t \right]p(\boldsymbol{z_t})d\boldsymbol{z_t}, \quad (6)$$

where $\boldsymbol{z_t}$ is the indicating vector consisting of binary elements, each element of which indicates whether the current task is relevant to the component of meta-knowledge or not. The multi-modal premise enables the sharing of meta-knowledge among different clusters of tasks via the overlapped related components, relaxing the restriction of mutual exclusiveness.

## 4.2 INDIAN BUFFET PROCESS PRIOR

In the non-stationary regime, one important requirement is to capture incremental information when a newer task is encountered. Thus, the fixed meta-knowledge is not appropriate. To capture the incremental meta-knowledge and fit the multi-modal premise, we employ the Indian Buffet Process (IBP) (Griffiths & Ghahramani, 2011) to make a prior decision on the number of components $\boldsymbol{z_t} \sim IBP(\alpha)$, where the number of the added components at each time is:

$$K_{t,new} \sim Possion(\frac{\alpha}{t}), \quad (7)$$

where $\alpha$ is the hyperparameter to control the rate of increase. The IBP prior for $\boldsymbol{z_t}$ is formulated based on the stick-breaking process:

$$v_k \sim Beta(\alpha, 1), \quad \pi_k = \prod_{i=1}^{k} v_i, \quad z_{t,k} \sim Bern(\pi_k), \quad for \ k = 1, ..., \infty. \quad (8)$$

where $Beta(\cdot)$ and $Bern(\cdot)$ represents the Beta distribution and the Bernoulli distribution, respectively. Based on the IBP prior, the generative process of SMM-CML is as follows:

$$\theta_{t,k} \sim \mathcal{N}(\mu_{t,k}, \sigma_{t,k}), \quad \phi_t|\theta_t, \boldsymbol{z}_t \sim p(\phi_t|\theta_t, \boldsymbol{z}_t), \quad (9)$$

where $\theta_{t,k}$ is the meta-knowledge of the $k$-th component, and the task-specific parameters $\phi_t$ are associated with a subset of meta-knowledge components determined by $\boldsymbol{z_t}$. The inference can be seen in Sec. 4.4 and the probability graph model is shown in Fig. 7 in the Appendix D.

Hereby the IBP provides a prior on the number of components when encountering new tasks so that it can capture the incremental knowledge in the online non-stationary setting. However, the IBP just provides a prior and it cannot make a posterior decision after the updating of meta-knowledge. When meeting a large number of task distributions, the unlimited increase in the number of components would cause a large computational consumption and lead to parameter inefficiency.

## 4.3 EVIDENTIAL SPARSIFICATION FOR MULTI-MODAL META-KNOWLEDGE

To learn the posterior number of components from tasks, we propose an evidential sparsification method for multi-modal meta-knowledge, which is a post hoc method after the update of meta-knowledge. Since the components in our multi-modal meta-knowledge are mutually independent, there might be redundancy across time. How to merge the information about different components from both previous and current times remains an issue. The evidential theory provides a good way to merge independent pieces of evidence and make the decision (Dempster, 2008).

After updating the meta-knowledge at one time, the relationship between the current clusters of tasks and each meta-knowledge component is built up. The relationship between tasks and one certain component is a piece of independent evidence, containing the support and doubt information. Such information from the current and previous times can be merged to illustrate the unified relationship between the occurring tasks and the meta-knowledge components. The components not receiving support are cast as redundant and removed. Fig. 6 in Appendix C shows an intuitive explanation of the combination between evidential theory and multi-modal meta-knowledge.

In our IBP-based meta-knowledge, the relationship between tasks and components at each time is determined by $k$ beta distributions of $v_{t,k}$. Following the evidential theory (Denœux, 2019), we see

each beta distribution at either the current time or the previous time as a piece of evidence, so that there are $t \cdot k$ pieces of evidence. Intuitively, as all beta distributions of $\boldsymbol{v}_{t,k}$ (i.e, the evidences) are independent, each of them only provides the support or doubt for the corresponding component. That is, it supports the corresponding component $\{z_k\}$ or the complementary set of the corresponding component $\overline{\{z_k\}}$. And this piece of evidence does not by itself provide 100% certainty, which in evidential theory means that the remaining probability commits to the universal set $Z$.

In this way, each piece of evidence (i.e., each beta distribution of $v_{t,k}$) can provide the evidential weight $w_{t,k}$, conducting two simple mass functions with the focal set $\{z_k\}$ and $\overline{\{z_k\}}$, respectively. The evidential weight can be defined as:

$$w_{t,k} = \exp(\alpha_{t,k}) - \gamma \exp(\beta_{t,k}), \tag{10}$$

where the evidential weight will increase with a larger $\alpha_{t,k}$ and decrease with a larger $\beta_{t,k}$. Note that the hyperparameter $\gamma$ can effectively adjust the sparsity of meta-knowledge. This weight $w_{t,k}$ can deduce two other evidential weights $w_{t,k}^+$ and $w_{t,k}^-$, supporting the singleton of corresponding component $\{z_k\}$ and its complementary set $\overline{\{z_k\}}$, respectively. The similar derivation as (Itkina et al., 2020) can be used as:

$$w_{t,k}^+ = max(0, w_{t,k}) > 0, \quad w_{t,k}^- = max(0, -w_{t,k}) > 0. \tag{11}$$

For each piece of evidence $v_{t,k}$, there exist two mass functions supporting $\{z_k\}$ and $\overline{\{z_k\}}$, respectively:

$$m_{t,k}^+(\{z_k\}) = 1 - exp(-w_{t,k}^+), \quad m_{t,k}^+(Z) = exp(-w_{t,k}^+); \tag{12}$$

$$m_{t,k}^-(\overline{\{z_k\}}) = 1 - exp(-w_{t,k}^-), \quad m_{t,k}^-(Z) = exp(-w_{t,k}^-). \tag{13}$$

And these mass functions provided by different pieces of evidence can be fused using Dempster's rule and get the final result as follows:

$$m(\{z_k\}) = CC^+C^- \left\{ exp(-w_k^-) \left[ exp(w_k^+) - 1 + \prod_{l \neq k}(1 - exp(-w_l^-)) \right] \right\}, \tag{14}$$

where $C, C^+$ and $C^-$ are the normalization terms and can be omitted when computing, and $w_k^+$ and $w_k^-$ are the merged evidential weight of each component. The computational details can be seen in Appendix C. If a component $k$ does not receive support (i.e., $m(z_k) = 0$), then there must be no evidence directly supporting this component (i.e., $w_k^+ = 0$) and there is at least one other component receiving doubt directly from the evidence (i.e., $w_l^- = 0, l \neq k$).

For the sparsity of the multi-modal meta-knowledge, we can apply the mass function developed above to filter out the component without receiving the support information directly. That is, components with zero singleton mass value (i.e., $m(\{z_k\}) = 0$) are removed and the construction of meta-knowledge for the specific task in Eq. 6 are modified as:

$$p(\phi_t|\theta_t, \boldsymbol{z}_t) = \sum_{k=1}^{K} \mathbb{1}\{m(\{z_k\}) \neq 0\} \mathbb{1}\{z_{t,k} \neq 0\} p(\phi_{t,k}|\theta_{t,k}; \lambda_{t,k}). \tag{15}$$

## 4.4 STRUCTURED VARIATIONAL INFERENCE

The exact inference is intractable because of non-conjugacy, thus, the approximation is required. In our work, we employ the variational inference (Blei et al., 2017) to approximate the posterior. The evidence lower bound (ELBO) of the observation at the current time $t$ can be derived as following:

$$\mathcal{L}(\psi_t, \mathcal{D}_t) = -\mathbb{E}_{q(v_t, \boldsymbol{z}_t, \theta_t, \phi_t)}[\log p(\mathcal{D}_t|\phi_t)] + \sum_{k=1}^{K} D_{KL}(q(v_{t,k})\|p(v_{t,k}))$$

$$+ \sum_{k=1}^{K} D_{KL}(q(z_{t,k}|v_{t,k})\|p(z_{t,k}|v_{t,k})) + \sum_{k=1}^{K} D_{KL}(q(\theta_{t,k})\|p(\theta_{t,k}))$$

$$+ D_{KL}(q(\phi_t|\theta_t, \boldsymbol{z}_t)\|p(\phi_t|\theta_t, \boldsymbol{z}_t)), \tag{16}$$

where $D_{KL}$ is the Kullback–Leibler divergence and $q(\cdot)$ is the variational distribution for each latent variables, respecively. Note that the expectation of likelihood in Eq. 16 can be computed using Monte Carlo sampling, while all the KL-terms can be computed directly as they have closed-form expressions via implicit reparameterization gradients (Figurnov et al., 2018). Details of the definition of variational distribution, the sampling gradient computation for the likelihood term and the closed form expression for KL-terms can be seen in Appendix D.

### 4.5 DISCUSSION

In contrast to recent works (Jerfel et al., 2019; Zhang et al., 2021), our work has two major differences that enhance the performance and confirm our contributions. (1) Different from the one-to-one matching between the cluster of tasks and the meta-knowledge component, our algorithm constructs a many-to-many matching, where multiple task clusters can share one meta-knowledge component and one task cluster needs multiple meta-knowledge components. This is to avoid bias toward one meta-knowledge component and improve performance on heterogeneous tasks. (2) Secondly, our algorithm combines the IBP prior with the evidential sparsification to learn the posterior number of meta-knowledge components. Compared to the existing works only using CPR as a prior, our algorithm makes a posterior decision, which achieves parameter efficiency and reduces computational consumption. The analysis of complexity is shown in Appendix E

## 5 EXPERIMENTS

To examine the effectiveness of our SMM-CML, we design experiments, make comparisons and analyze the results. Specifically, the research problems that guide the remainder of the paper are: **(RQ1)** Can our proposed SMM-CML achieve a better performance than the state-of-the-art baselines under the online non-stationary setting? **(RQ2)** Can the increasing number of components capture the incremental information? **(RQ3)** What is the impact of evidential sparsification on performance?

Our experiments are conducted under the online non-stationary settings. We compare our algorithm to the following baselines: (1) **Train-On-Everything (TOE)**: an intuitive method that re-initializes the meta-knowledge at each time $t$ and trains on all the arriving data $\mathcal{D}_{1:t}$; (2) **Train-From-Scratch (TFS)**: another intuitive method that also re-initializes the meta-knowledge at each time $t$ but trains only on the current data $\mathcal{D}_t$; (3) **Follow the Meta Leader (FTML)**(Finn et al., 2019): a method utilizing the Follow the Leader algorithm (Kalai & Vempala, 2005) to minimize the regret of meta-learner. (4) **Online Structured Meta-Learning (OSML)**: a method via conducting a pathway to extract meta-knowledge from a meta-hierarchical graph; (5) **Dirichlet Process Mixture Model (DPMM)**: an algorithm that employs CRP to conduct a mixture meta-knowledge using point estimation; (6) **Bayesian Online Meta-Learning with Variational Inference (BOMVI)**: a method that uses Bayesian meta-learning to address the catastrophic forgetting issue; (7) **Variational Continual Bayesian Meta-Learning (VC-BML)**: a state-of-the-art method that aims to conduct a mixture meta-knowledge via a Bayesian method.

Following the exiting works (Yap et al., 2021; Zhang et al., 2021), we conduct the experiments on four datasets: *VGG-Flowers*(Nilsback & Zisserman, 2008), *miniImagenet*(Ravi & Larochelle, 2017), *CIFAR-FS*(Bertinetto et al., 2018), and *Omniglot*(Lake et al., 2011). Tasks sampled from different datasets correspond to different task distribution, so that the online non-stationary environment can be created via chronologically sampling tasks from different datasets. Specifically, the sampled task is a 5-way 5-shot task, and 5 classes are sampled randomly from a dataset for a task. In our experiment, we sequentially meta-train the model on tasks sampled from the meta-training dataset of these four datasets, which means that the model is trained on the tasks sampled from *VGG-Flowers* dataset, and then proceeds to the next dataset. The showed performances are evaluated on the test set after tuning hyper-parameters on the validation set. More details about experiment are in Appendix F.

### 5.1 RQ1: PERFORMANCE UNDER ONLINE NON-STATIONARY SETTING

To examine the effectiveness of our algorithm, we present the mean meta-test accuracy on all the learned datasets at each meta-training stage in Tab. 1, and the details of performance on each training stage are in Appendix F.4.1. Our SMM-CML achieves the best performance at each meta-training stage (i.e., *VGG-Flowers*, *miniImagenet*, *CIFAR-FS* and *Omniglot*), which illustrates

Table 1: Mean meta-test accuracy (%) of the learned dataset at each meta-training stage. The best performance is marked with boldface.

| Algorithms | VGG-Flowers | miniImagenet | CIFAR-FS | Omniglot |
|---|---|---|---|---|
| FTML | $76.84 \pm 1.75$ | $60.74 \pm 1.85$ | $66.71 \pm 1.86$ | $61.89 \pm 1.49$ |
| OSML | $79.61 \pm 1.50$ | $66.15 \pm 1.73$ | $68.24 \pm 1.73$ | $65.65 \pm 1.40$ |
| DPMM | $78.97 \pm 1.52$ | $66.55 \pm 1.77$ | $67.18 \pm 1.86$ | $68.26 \pm 1.47$ |
| BOMVI | $77.05 \pm 1.80$ | $60.44 \pm 1.86$ | $59.57 \pm 1.77$ | $69.04 \pm 1.54$ |
| VC-BML | $83.71 \pm 1.58$ | $68.09 \pm 1.58$ | $69.87 \pm 1.74$ | $69.48 \pm 1.51$ |
| SMM-CML | $\mathbf{85.11 \pm 1.46}$ | $\mathbf{69.45 \pm 1.54}$ | $\mathbf{70.72 \pm 1.61}$ | $\mathbf{71.46 \pm 1.39}$ |

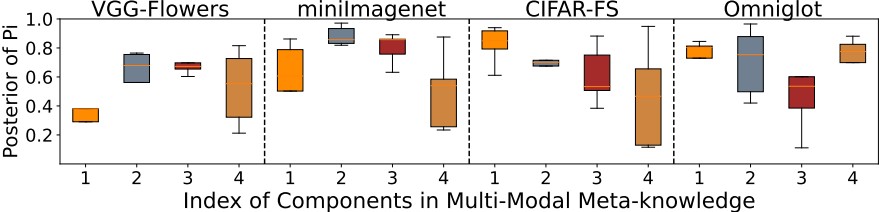

Figure 3: Each column represents the posterior probability $\pi$ of the Bernoulli distribution of different component in the multi-modal meta-knowledge on different datasets.

Table 2: Mean meta-test accuracy (%) under the sequential task setting, where the performance represents the average accuracy across the whole training tasks sequence. The best performance is marked with boldface.

| Algorithms | VGG-Flowers | miniImagenet | CIFAR-FS | Omniglot |
|---|---|---|---|---|
| FTML | $57.29 \pm 2.28$ | $31.92 \pm 1.58$ | $39.21 \pm 1.75$ | $82.03 \pm 1.42$ |
| OSML | $56.07 \pm 2.10$ | $32.41 \pm 1,36$ | $40.75 \pm 1.85$ | $82.89 \pm 1.42$ |
| DPMM | $64.21 \pm 2.06$ | $36.68 \pm 1.46$ | $47.47 \pm 1.88$ | $88.39 \pm 1.48$ |
| BOMVI | $64.71 \pm 1.78$ | $38.44 \pm 1.41$ | $48.19 \pm 1.88$ | $90.49 \pm 1.62$ |
| VC-BML | $65.28 \pm 2.19$ | $38.65 \pm 1.83$ | $47.07 \pm 1.75$ | $89.97 \pm 1.11$ |
| SMM-CML | $\mathbf{66.27 \pm 2.01}$ | $\mathbf{40.04 \pm 1.48}$ | $\mathbf{48.97 \pm 1.71}$ | $\mathbf{91.13 \pm 0.20}$ |

that SMM-CML can not only maintain the meta-knowledge learned from previous times but also capture the incremental meta-knowledge from the current tasks. Moreover, the comparison between the performance of SMM-CML and the baselines (i.e., DPMM and VC-CML), which maintain the mutually exclusive meta-knowledge components, confirms that sharing helps to improve performance.

To further illustrate the association between tasks and meta-knowledge, we show the posterior of the Bernoulli distribution of each component on each dataset. As in Fig. 3, the probabilities of Bernoulli distribution of each component are distinct. For example, the *VGG-Flowers* dataset has a strong association with the later three components, while the *Omniglot* dataset is closely relevant to all the components except the third one. It confirms that in our learned meta-knowledge, different clusters of tasks share multiple components and maintain their diversity via the other different ones.

Besides, we take into account another more challenging setting, where tasks from different datasets are mixed and randomly arrive one by one. Because of the more non-stationary task stream, catastrophic forgetting is more serious. Tab. 2 shows the average accuracy results over all times. Our SMM-CML achieves the best performance on all four datasets even in such a challenging setting. It further confirms that our algorithm has the capability to cope with the online non-stationary task streams.

## 5.2 RQ2: The Impact of Increasing Number of Components

To capture the incremental meta-knowledge in continual meta-learning, we employ the Indian Buffet Process to allow the increasing number of components. We conduct the experiment with different numbers of meta-knowledge components to test its effectiveness. The evolution of meta-test accuracy when training on different datasets is shown in Fig. 4. TOE has the best performance on most stages because it can replay all the available data. With the number of components increasing, our proposed algorithm has a better performance in both the learned and the new datasets. It further demonstrates that more components can capture the incremental meta-knowledge and alleviate the forgetting issue.

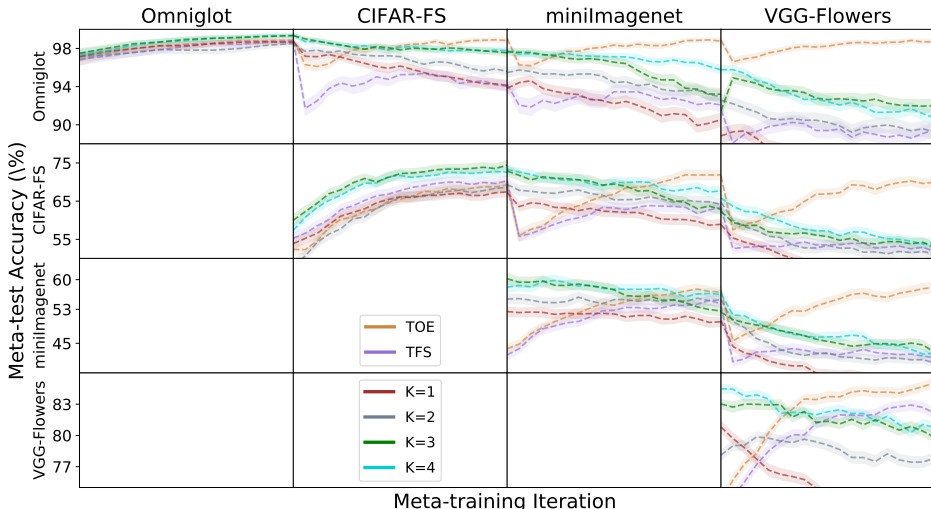

Figure 4: The evolution of meta-test accuracies (%) of SMM-CML with different numbers of components when training on different datasets. TOE and TFS are two baselines for comparison.

Table 3: The meta-test accuracy (%) before sparsification and after sparsification on each dataset

|  | **Omniglot** | **CIFAR-FS** | **miniImagenet** | **VGG-Flowers** |
|---|---|---|---|---|
| original | $99.31 \pm 0.25$ | $85.99 \pm 1.07$ | $76.07 \pm 1.40$ | $71.41 \pm 1.48$ |
| sparse | $99.43 \pm 0.21$ | $86.53 \pm 1.05$ | $75.91 \pm 1.38$ | $71.22 \pm 1.36$ |

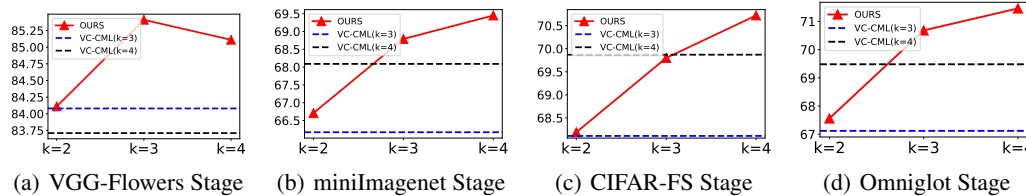

(a) VGG-Flowers Stage  (b) miniImagenet Stage  (c) CIFAR-FS Stage  (d) Omniglot Stage

Figure 5: The comparison between SMM-CML and VC-BML with different numbers of components on each training stage.

## 5.3 RQ3: THE EFFECTIVENESS OF EVIDENTIAL SPECIFICATION

To reduce computational consumption, we propose an evidential sparsification method. To examine the impact of our methods, we compare performance before and after sparsification. The mean meta-test accuracy at each meta-training stage is shown in Tab. 3, and more results are shown in Appendix F.4.2. Compared to the original meta-knowledge, the sparse meta-knowledge can achieve a comparative performance. This confirms that our method can reduce redundancy and computational consumption with acceptable accuracy. Moreover, we conduct experiments on different numbers of components with the appropriate $\gamma$. The results in Fig. 5 show that our model can outperform the SOTA even with less number of components. It confirms that our algorithm can filter out the redundant meta-knowledge component and is more parameter-efficiency.

## 6 CONCLUSION

This paper focuses on a more challenging setting in meta-learning, where tasks from a non-stationary distribution are available sequentially. We propose SMM-CML, a Scalable Multi-Modal Meta-Learning algorithm where a cluster of similar tasks are associated with multiple components, allowing tasks to share meta-knowledge while maintaining their diversity. Moreover, an IBP prior is employed to determine whether to increase the number of components, and an evidential sparsity method is proposed to filter out the components which have not received support information from tasks. This confirms a posterior number of meta-knowledge components so that it avoids parameter inefficiency. The conducted experiment shows the effectiveness of multi-modal meta-knowledge and confirms that our algorithm can learn the needed meta-knowledge from tasks. One limitation comes from the space complexity, since our model still needs to increase the number of mixture components to cover more meta-knowledge. The proposed evidential sparsity method can help alleviate the required space complexity.

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

## A    VARIATIONAL INFERENCE FOR META-LEARNING

Following MAML (Finn et al., 2017), many Bayesian variant (Ravi & Beatson, 2018; Gordon et al., 2019; Iakovleva et al., 2020) are proposed. To fit well with the bi-level optimization architecture, most of them consider a hierarchical bayesian inference (Amit & Meir, 2018), where the Evidence Lower Bound (ELBO) of likelihood can be derived as follows:

$$
\log\left[\prod_{i=1}^{T}\mathcal{D}_i\right] = \log\left[\int p(\theta)\left[\prod_{i=1}^{T}\int p(\mathcal{D}_i|\phi_i)p(\phi_i|\theta)d\phi_i\right]d\theta\right]
$$

$$
\geq \mathbb{E}_{q(\theta;\psi)}\left[\log\left(\prod_{i=1}^{T}\int p(\mathcal{D}_i|\phi_i)p(\phi_i|\theta)d\phi_i\right)\right] - D_{KL}(q(\theta;\psi)||p(\theta))
$$

$$
\geq \mathbb{E}_{q(\theta;\psi)}\left[\sum_{i=1}^{T}\mathbb{E}_{q(\phi_i;\lambda_i)}\left[\log p(\mathcal{D}_i|\phi_i) - D_{KL}(q(\phi_i;\lambda)||p(\phi_i|\theta))\right]\right] - D_{KL}(q(\theta;\psi)||p(\theta)), \quad (17)
$$

where $\theta$ and $\phi$ are the global parameter and task-specific parameter, respectively. Note that the low bound is derived based on the Jensen equation and the variational distributions of $\theta$ and $\phi$ are introduced to approximate the intractable posterior. Then, the bi-level optimization is transformed as:

$$
\phi^*, \lambda^* = \underset{\psi,\lambda}{\arg\max}\, \mathbb{E}_{q(\theta;\psi)}\left[\sum_{i=1}^{T}\mathbb{E}_{q(\phi_i;\lambda_i)}\left[\log p(\mathcal{D}_i|\phi_i) - D_{KL}(q(\phi_i;\lambda)||p(\phi_i|\theta))\right]\right]
$$
$$
- D_{KL}(q(\theta;\psi)||p(\theta)). \quad (18)
$$

So that the goal of the optimization is to seek the optimal variational distribution of $\theta$ and $\phi$, parameterized by $\psi$ and $\lambda$, respectively.

## B    COMPUTING RULES IN EVIDENTIAL THEORY

There are some computing rules introduced by Dempster–Shafer theory (Denœux, 2019). Given two mass functions $m_1$ and $m_2$, their combination is defined according to the Dempster's rule:

$$
(m_1 \oplus m_2)(A) = \frac{1}{1-\kappa}\sum_{B\cap C=A} m_1(B)\cdot m_2(C), \quad (19)
$$

where $\kappa$ is the degree of conflict between two evidences, which is defined as:

$$
\kappa = \sum_{B\cap A=\emptyset} m_1(B)\cdot m_2(C). \quad (20)
$$

Note that Dempster's rule for the combination of mass functions is commutative and associative. Based on Dempster's rule for the combination between two mass functions, the combination of two corresponding contour functions $pl_1$ and $pl_2$ can be computed as:

$$
(pl_1 \oplus pl_2(z_k)) = \frac{pl_1(z_k)\cdot pl_2(z_k)}{1-\kappa}. \quad (21)
$$

And if both mass functions are simple with the same strict subset, their fusion can be defined as:

$$
A^{w_1} \oplus A^{w_1} = A^{w_1+w_2}, \quad (22)
$$

where $A^{w_1}$ represents the simple mass function with a single strict subset and its evidential weight is $w_1$.

## C    THE COMPUTATIONAL DETAILS OF FUSING MASS FUNCTION

We try to combine all the positive mass functions and all the negative mass functions, respectively. And then the two can be fused to produce the final result.

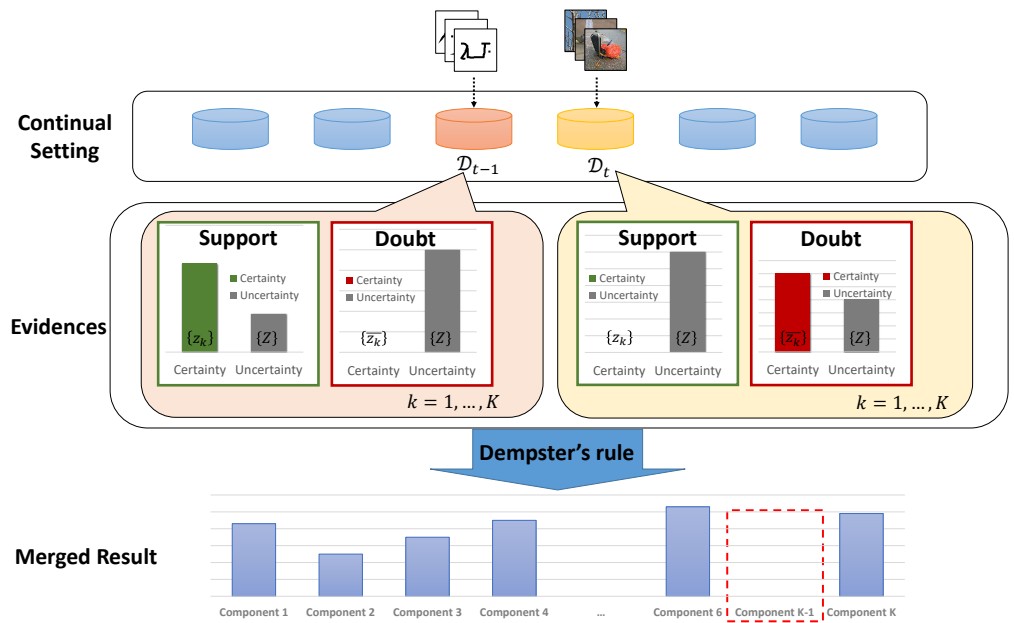

Figure 6: An intuitive explanation of our proposed evidential sparsity method. The relationship between tasks and meta-knowledge components at each time provides $k$ evidences, containing support and doubt information with uncertainty. Such information can be merged by Dempster's rule to provide a unified relationship between the occurring tasks and the components. The components not receiving support are removed (i.e., the component $K - 1$ in the figure).

## C.1 THE FUSION ACROSS TIME

Before positive fusion and negative fusion, we need to merge evidence supporting the same focal elements at different times. Since the simple mass functions have the same focal set, their fusion can be calculated following Eq. 22 and the weight is:

$$w_k^+ = \sum_{i=0}^{t} w_{i,k}^+, \quad w_k^- = \sum_{i=0}^{t} w_{i,k}^- \tag{23}$$

where $w_{i,k}^+$ and $w_{i,k}^-$ are the evidential weight of the positive and negative mass function at time $i$. respectively. In this way, the evidence supporting the same focal element from different time can be merged first:

$$m_k^+(\{z_k\}) = 1 - exp(-w_k^+), \quad m_k^+(Z) = exp(-w_k^+); \tag{24}$$

$$m_k^-(\overline{\{z_k\}}) = 1 - exp(-w_k^-), \quad m_k^-(Z) = exp(-w_k^-). \tag{25}$$

## C.2 THE FUSION OF $m^+$

As we define above, all the positive mass functions have the only two focal elements, $\{z_k\}$ and $Z$. Then the combination of them can be computed according to the Dempster's rule:

$$m^+(\{z_k\}) \propto [1 - exp(-w_k^+)] \prod_{l \neq k} exp(-w_k^+) = [exp(w_k^+) - 1] \prod_{l=1}^{K} exp(-w_k^+), \tag{26}$$

$$m^+(Z) \propto \prod_{k=1}^{K} exp(-w_k^+). \tag{27}$$

As the fused mass function constraint to the sum of one, the results can be computed by normalizing the terms. So that the sum of all terms is:

$$m^+(Z) + \prod_{l=1}^{K} m^+(\{z_k\}) \propto \left( \prod_{k=1}^{K} exp(-w_k^+) \right) + \sum_{k=1}^{K} \left\{ [exp(w_k^+) - 1] \prod_{l=1}^{K} exp(-w_k^+) \right\} \quad (28)$$

$$= \left( \prod_{k=1}^{K} exp(-w_k^+) \right) \cdot \left[ \left( \sum_{k=1}^{K} exp(w_k^+) \right) - K + 1 \right]. \quad (29)$$

And the terms can be normalized as:

$$m^+(\{z_k\}) = \frac{[exp(w_k^+) - 1] \prod_{l=1}^{K} exp(-w_k^+)}{\left( \prod_{k=1}^{K} exp(-w_k^+) \right) \cdot \left[ \left( \sum_{k=1}^{K} exp(w_k^+) \right) - K + 1 \right]}$$

$$= \frac{exp(w_k^+) - 1}{\left( \sum_{k=1}^{K} exp(w_k^+) \right) - K + 1}, \quad (30)$$

$$m^+(Z) = \frac{\prod_{k=1}^{K} exp(-w_k^+)}{\left( \prod_{k=1}^{K} exp(-w_k^+) \right) \cdot \left[ \left( \sum_{k=1}^{K} exp(w_k^+) \right) - K + 1 \right]}$$

$$= \frac{1}{\left( \sum_{k=1}^{K} exp(w_k^+) \right) - K + 1}. \quad (31)$$

### C.3 THE FUSION OF $m^-$

Different from the positive mass functions, the negative mass functions have the only two focal elements, $\overline{\{z_k\}}$ and $Z$. To compute the combination of all negative mass functions, we need to compute the conflict firstly:

$$\kappa^- = \prod_{k=1}^{K} \left( 1 - exp(-w_k^-) \right). \quad (32)$$

Thus, for any strict subset $A$ of $Z$, its belief can be computed as:

$$m^-(A) = \frac{\left[ \prod_{z_k \notin A} \left( 1 - exp(-w_k^-) \right) \right] \cdot \left[ \prod_{z_k \in A} exp(-w_k^-) \right]}{1 - \prod_{k=1}^{K} \left( 1 - exp(-w_k^-) \right)}. \quad (33)$$

And the mass belief of the complete set $Z$ is:

$$m^-(Z) = \frac{\prod_{k=1}^{K} exp(-w_k^-)}{1 - \prod_{k=1}^{K} \left( 1 - exp(-w_k^-) \right)}. \quad (34)$$

For further fusion of the positive and negative mass functions, we need to compute $pl^-(z_k)$, which can be defined as:

$$pl^-(\{z_k\}) = \frac{\prod_{k=1}^{K} pl_k^-(\{z_k\})}{1 - \prod_{k=1}^{K} \left( 1 - exp(-w_k^-) \right)}, \quad (35)$$

where the plausibility of negative mass function is:

$$pl_l^-(\{z_k\}) = \begin{cases} exp(-w_l^-) & if \ k = l \\ 1 & otherwise \end{cases}. \quad (36)$$

Thus, the result of the fused plausibility is:

$$pl^-(\{z_k\}) = \frac{exp(-w_k^-)}{1 - \prod_{k=1}^{K} \left( 1 - exp(-w_k^-) \right)}. \quad (37)$$

## C.4 THE FINAL FUSION

To clarify the following derivation, we assume that:

$$C^+ = \frac{1}{\left(\sum_{k=1}^{K} exp(w_k^+)\right) - K + 1}, \tag{38}$$

$$C^- = \frac{1}{1 - \prod_{k=1}^{K} \left(1 - exp(-w_k^-)\right)}. \tag{39}$$

Similarly, to combine the positive and negative mass function, we need to compute the conflict between them at first:

$$\kappa = \sum_{k=1}^{K} \left\{ m^+(\{z_k\}) \left[ \sum_{z_k \notin A} m^-(A) \right] \right\}$$

$$= \sum_{k=1}^{K} \left\{ m^+(\{z_k\}) \cdot \left[ 1 - pl^-(\{z_k\}) \right] \right\}$$

$$= \sum_{k=1}^{K} \left\{ C^+ \left[ exp(w_k^+) - 1 \right] \cdot \left[ 1 - C^-(exp(-w_k^-)) \right] \right\}, \tag{40}$$

where $A \subseteq Z$. To make the following derivation clarified, let:

$$C = \frac{1}{1 - \kappa} = \frac{1}{1 - \sum_{k=1}^{K} \left\{ C^+ \left[ exp(w_k^+) - 1 \right] \cdot \left[ 1 - C^-(exp(-w_k^-)) \right] \right\}}. \tag{41}$$

Then for any $k \in \{1, 2, ..., K\}$, the mass belief of each singleton can be computed as:

$$m(\{z_k\}) = C \left\{ m^+(\{z_k\}) \cdot \left[ \sum_{z_k \in A} m^-(A) \right] + m^+(Z) \cdot m^-(\{z_k\}) \right\}$$

$$= C \left\{ m^+(\{z_k\}) \cdot pl^-(\{z_k\}) + m^+(Z) \cdot m^-(\{z_k\}) \right\}, \tag{42}$$

where $A \subseteq Z$. Combining Eq. 30, Eq. 31 Eq. 33 and Eq. 37, , the final result of the mass singleton belief is:

$$m(\{z_k\})$$

$$= C \left\{ C^+ \left[ exp(w_k^+) - 1 \right] \cdot C^- [exp(-w_k^-)] + C^+ \cdot C^- \left[ exp(-w_k^-) \cdot \prod_{l \neq k} \left( 1 - exp(-w_l^-) \right) \right] \right\}$$

$$= CC^+C^- \left\{ exp(-w_k^-) \left[ exp(w_k^+) - 1 + \prod_{l \neq k} (1 - exp(-w_l^-)) \right] \right\}. \tag{43}$$

# D    DETAILS OF INFERENCE

In this section, we present the details of our structured variational inference for our proposed SMM-CML. The pseudo-code and the probability model are shown in Alg.1 and Fig. 7, respectively.

## D.1    VARIATIONAL DISTRIBUTION

Because of the intractability of posterior, we introduce the variational distribution to approximate the true posterior. To capture the dependencies among the approximate posterior distribution, we consider using the structured mean-field approximation (Hoffman & Blei, 2015) instead of the traditional mean-field approximation. Specifically, the joint variational distribution can be decomposed as follows:

$$q(v_t, \boldsymbol{z}_t, \theta_t, \phi_t | \mathcal{D}_t) = q(\phi_t | \theta_t, \boldsymbol{z}_t, \mathcal{D}_t) \prod_{k=1}^{K} q(\theta_{t,k}) q(z_{t,k} | v_{t,k}) q(v_{t,k}), \tag{44}$$

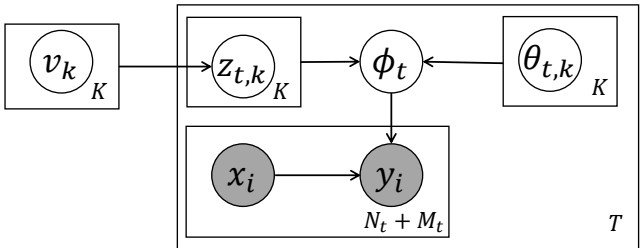

Figure 7: The probability model of SMM-CML. The solid line denotes the generative process, and the white circle and the grey circle denote the latent variant and the observed variant, respectively.

The composed variational distributions are parameterized as:

$$q(v_{t,k}) = Beta(\alpha_{t,k}, \beta_{t,k}), \tag{45}$$

$$q(z_{t,k}|\pi_{t,k}) = Bern(\pi_{t,k}), \quad where \ \pi_{t,k} = v_{t,k}, \tag{46}$$

$$q(\theta_{t,k}) = \mathcal{N}(\mu_{t,k}, \sigma_{t,k}^2 \mathbb{1}), \tag{47}$$

$$q(\phi_t|\theta_t, \boldsymbol{z}_t, \mathcal{D}_t) = \sum_{k=1}^{K} \mathbb{1}\{z_{t,k} \neq 0\} q(\phi_{t,k}|\theta_{t,k}; \lambda_{t,k}), \tag{48}$$

where $\lambda_t = SGD_J(\theta_t^*, \mathcal{D}_t^S, \epsilon)$, and $SGD_J(\cdot)$ represents the stochastic gradient descent with J steps. That is, the required variational parameters are $\psi_t = \{\alpha_{t,k}, \beta_{t,k}, \mu_{t,k}, \sigma_{t,k}, \lambda_{t,k}\}$ for all $k = 1, ..., K$. Note that we replace $\prod_{i=1}^{k} v_{t,i}$ with $v_{t,k}$ in the posterior, to remove the implicit order constraint in the prior. So that the optimization aims to search for the optimal variational parameter to maximize the ELBO in Eq. 16.

### D.2 Reparameterization

The variational posterior is obtained by optimizing the ELBO using structured variational inference. To make inference tractable, we utilize three reparameterizations, to infer the Gaussian distribution, beta distribution and Bernoulli distribution, respectively.

#### D.2.1 The variational Gaussian distribution reparameterization

As we mentioned above, the variational distributions of meta-knowledge from each clusters are diagonal Gaussian $\theta_{t,k} \sim \mathcal{N}(\mu_{t,k}, \sigma_{t,k})$. We employ the reparameterization, which can represent the meta-knowledge using a deterministic function $\theta_{t,k} = g(\varepsilon; \mu_{t,k}, \sigma_{t,k})$, where $\epsilon \sim \mathcal{N}(0, I)$. To apply the reparameterization, we define the standardization function and its inverse as:

$$\mathcal{S}_\psi(\theta) = \frac{\theta - \mu}{\sigma} = \varepsilon \sim q(\varepsilon), \ \ where \ q(\varepsilon) = \mathcal{N}(0, I),$$
$$\theta = \mathcal{S}_\phi^{-1}(\varepsilon) = \varepsilon \cdot \sigma + \mu. \tag{49}$$

Note that we omit the subscripts for clarity and the remainder of this section omits them as well. Then we can represent the objective in ELBO w.r.t $q(\theta)$ as follows:

$$\mathbb{E}_{q_\psi(\theta)}[f(\theta)] = \mathbb{E}_{q(\varepsilon)}[f(\mathcal{S}_\psi^{-1}(\varepsilon))]. \tag{50}$$

This allows us to compute the gradient of the expectation in another way:

$$\nabla_\psi \mathbb{E}_{q_\psi(\theta)}[f(\theta)] = \mathbb{E}_{q(\varepsilon)}[\nabla_\psi f(\mathcal{S}_\psi^{-1}(\varepsilon))] = \mathbb{E}_{q(\varepsilon)}[\nabla_\theta f(\mathcal{S}_\psi^{-1}(\varepsilon))\nabla_\psi \mathcal{S}_\psi^{-1}(\varepsilon)], \tag{51}$$

#### D.2.2 The variational Beta distribution reparameterization

There is no simple inverse of the standardization function when using the reparameterization for Beta distribution, which makes it impossible to apply the explicit reparameterization directly. Instead,

---

**Algorithm 1:** The meta-training process of SMM-CML.

---

**Input:** Task distribution $p(\tau)$, data distribution $p(\mathcal{D}|\tau)$,
the initial number of component $K_0$, concentration parameter $\alpha$,
the number of inner update step $J$, the inner learning rate $\epsilon$,
and the outer learning rate $\zeta$

1: **for** t=1,.. **do**
2:     Determine the added number: $K_{t,new} = Possion(\frac{\alpha}{t})$
3:     Determine the number of component: $K_t = K_{t-1} + K_{t,new}$
4:     Initialize the variational beta distribution: $\alpha_{t,k}, \beta_{t,k}, \forall k = 1, ..., K_t$
5:     Initialize the variational distribution of meta-knowledge: $\mu_k, \sigma_k, \forall k = 1, .., K_t$
6:     **while** not converge **do**
7:         Sample $v_{t,k} \sim q(v_{t,k}; \alpha_{t,k}, \beta_{t,k}), \forall k = 1, ..., K_t$
8:         Compute the ELBO according to Eq. 16
9:         Compute the gradient: $\nabla\mu_{t,k}, \nabla\sigma_{t,k}, \forall k = 1, ..., K_t$ via explicit reparameterization
           according to Eq. 51
10:       Compute the gradient: $\nabla\alpha_{t,k}, \nabla\beta t, k, \forall k = 1, ..., K_t$ via implicit reparameterization
           according to Eq. 55
11:       Update the variational parameters: $\alpha_{t,k} \leftarrow \alpha_{t,k} - \zeta\nabla\alpha_{t,k}, \forall k = 1, ..., K_t$
12:       Update the variational parameters: $\beta_{t,k} \leftarrow \beta_{t,k} - \zeta\nabla\beta_{t,k}, \forall k = 1, ..., K_t$
13:       Update the variational parameters: $\mu_{t,k} \leftarrow \mu_{t,k} - \zeta\nabla\mu_{t,k}, \forall k = 1, ..., K_t$
14:       Update the variational parameters: $\sigma_{t,k} \leftarrow \sigma_{t,k} - \zeta\nabla\sigma_{t,k}, \forall k = 1, ..., K_t$
15:     **end while**
16:     Update prior: $p(v_{t,k}) \leftarrow q(v_{t,k}), \forall k = 1, ..., K_t$
17:     Compute the evidential weight $w_{t,k}^+, w_{t,k}^-, \forall k = 1, ..., K_t$ according to Eq. 11
18:     Compute the mass function according to Eq. 14
19:     Remove the components without support information according to Eq. 15
20: **end for**

---

there are two ways to tackle the problem: the implicit reparameterization and the Kumaraswamy reparameterization.

**Implicit reparameterization.** This way also utilizes the reparameterization to tackle the intractable gradient in Beta distribution:

$$\nabla_\gamma \mathbb{E}_{q_\gamma(v_k)}[f(v_k)] = \mathbb{E}_{q(\varepsilon)}[\nabla_\gamma f(v_k)] = \mathbb{E}_{q(\varepsilon)}[\nabla_{v_k} f(v_k)\nabla_\gamma v_k], \tag{52}$$

without the inverse of the standardization function, the term $\nabla_\gamma v_k$ is difficult to compute. Inspired by (Figurnov et al., 2018), we employ the implicit reparameterization to compute the gradient, the idea of which is to differentiate the standardization function $\mathcal{S}_\gamma(v_k) = \varepsilon$ using the chain rule instead of searching its inverse:

$$\nabla_{v_k}\mathcal{S}_\gamma(v_k)\nabla_\gamma(v_k) + \nabla_\gamma\mathcal{S}_\gamma(v_k) = \mathbf{0}, \tag{53}$$

$$\nabla_\gamma v_k = -(\nabla_{v_k}\mathcal{S}_\gamma(v_k))^{-1}\nabla_\gamma\mathcal{S}_\gamma(v_k). \tag{54}$$

Note that the standardization function can be the CDF of the Beta distribution and $\varepsilon \sim Unif[0, 1]$. Then the implicit gradient is:

$$\nabla_\gamma v_k = \frac{\nabla_\gamma F(v_k; \gamma)}{-(\nabla_{v_k} F(v_k; \gamma))} = \frac{\nabla_\gamma F(v_k; \gamma)}{-p(v_k; \gamma)}, \tag{55}$$

where $p(v_k; \gamma)$ is the PDF of the Beta distribution.

**Kumaraswamy distribution.** The Beta distribution of $v_k$ also can be reparameterized using a Kumaraswamy distribution (Nalisnick & Smyth, 2017). The Kumaraswamy distribution can be defined as:

$$p(v_k; \alpha, \beta) = \alpha\beta v_k^{\alpha-1}(1 - v_k^\alpha)^{\beta-1}, \tag{56}$$

and then the inverse of standardization function can be computed as:

$$\mathcal{S}_\gamma(v_k) = (1 - \varepsilon^{1/\beta})^{1/\alpha}, \ \ where \ \varepsilon \sim Unif[0, 1]. \tag{57}$$

The KL-Divergence between the Kumaraswamy distribution and the Beta distribution in ELBO can be written as:

$$
\begin{aligned}
D_{KL}\left(q(v_k;\alpha_k,\beta_k)\|p(v;\alpha,\beta)\right) = & \frac{\alpha_k-\alpha}{\alpha_k}\left(-\gamma-\Psi(\beta_k)-\frac{1}{\beta_k}\right)+\log\alpha_k\beta_k \\
& + \log\left[B(\alpha,\beta)\right]-\frac{\beta_k}{1-\beta_k} \\
& + (\beta-1)\beta_k\sum_{m=1}^{\infty}\frac{1}{m+\alpha_k\beta_k}B\left(\frac{m}{\alpha_k},\beta_k\right),
\end{aligned}
\tag{58}
$$

where $\gamma$ is the Euler constant, $\Psi(\cdot)$ is the digamma function, and $B(\cdot,\cdot)$ is the beta function. Following the existing work (Nalisnick & Smyth, 2017), the above the infinite term in the formula can be approximated using a infinite sum of the first 11 terms.

### D.2.3 THE VARIATIONAL BERNOULLI DISTRIBUTION REPARAMETERIZATION

As the Bernoulli distribution is one of the classic discrete distributions, the sampling requires performing an $argmax$ operation. But the $argmax$ operation is not differentiable.

We employ the Concrete distribution (Maddison et al., 2017), also named Gumbel-softmax distribution (Jang et al., 2017), to address the above issue. Then, we can sample a random variable as follows:

$$
x_j = \sigma\left(\frac{\log(\pi_k)+\log\left(\frac{u_k}{1-u_k}\right)}{\lambda}\right), \quad u \sim U(0,1),
\tag{59}
$$

where $\lambda \in (0,\infty)$ is a temperature hyper-parameter, $\sigma(\cdot)$ is the sigmoid function, $\pi_k$ is the parameter of the Bernoulli distribution and $u_k$ is sampled from a uniform distribution $U$. To guarantee a lower bound on the ELBO, both posterior and prior Bernoulli distribution need to be replaced with concrete distribution:

$$
D_{KL}\left[q(\mathbf{z}_t|\pi_{k,t})\|p(\mathbf{z}_t|\pi_{k,t})\right] \geq D_{KL}\left[q(\mathbf{z}_t|\pi_{k,t},\lambda)\|p(\mathbf{z}_t|\pi_{k,t},\lambda)\right].
\tag{60}
$$

## E  THE ANALYSIS OF COMPLEXITY

We discuss the computational cost of our proposed SMM-CML as follows, including the time complexity and space complexity.

For time complexity, the *de facto* bi-level optimization mechanism in meta-learning requires $O(n^2)$ when updating one meta-knowledge component, where an algorithm with time complexity $O(n)$ is a linear time algorithm. If without any sparsification or constraint on the number of components, it will see an unlimited increase, and thus the time complexity will be up to $O(n^3)$. If with our evidential sparsification, the number of components will be limited to a small constant $C$ with an appropriate hyperparameter $\gamma$ in Eq. 10, so that the time complexity will be down to $O(C*n^2) \approx O(n^2)$.

Similarly, as each component of meta-knowledge contains the parameter of the model, its space complexity is $O(n)$. And the total space complexity of models without sparsification will be up to $O(n^2)$ for the unlimited number of meta-knowledge components when encountering many tasks. But our algorithm can alleviate this issue using the evidential sparsification to reduce down to $O(n)$ with an appropriate hyperparameter $\gamma$.

## F  DETAILS OF EXPERIMENT

### F.1  THE DETAILS OF BASELINES

For a fair comparison, we use the widely-applied network architecture following (Yap et al., 2021; Zhang et al., 2021). In what follows, we describe the details of the baselines:

**TOE**: Training-On-Everything method (TOE) is an intuitive method, that re-initializes the meta-knowledge and trains them on all the having arrived datasets at each time. We use the same Bayesian

meta-learning architecture as our algorithm. The difference between TOE and SMM-CML is that SMM-CML is only trained on the current dataset at each time instead of all the having arrived dataset in TOE and SMM-CML does not re-initialize the meta-knowledge at each time as what TOE do.

**TFS**: Train-From-Scratch (TFS) is another intuitive method, which also re-initializes meta-knowledge but only trains them on the current dataset. Similarly, it also uses the same Bayesian meta-learning architecture as our algorithm. The difference between TFS and SMM-CML is that our algorithm maintains the posterior meta-knowledge at last time as the prior at the current time instead of re-initializing them as TFS.

**FTML**: Follow the Meta Leader (FTML) proposed by (Finn et al., 2019) uses the Follow the Leader algorithm to fill the gap between meta-learning and online learning. However, it assumes that all the having arrived datasets are available, which is memory-consuming and conflicts with the continual meta-learning. For a fair comparison, we only train FTML on the current dataset as same as our algorithm.

**OSML**: Online Structured Meta-Learning (OSML) (Yao et al., 2020) maintains a meta-hierarchical graph with different knowledge blocks and conducts a meta-knowledge pathway for the encountered new task. However, it employs a well pre-trained convolution network to initialize the model in the original paper. As SMM-CML and other baselines are randomly initialized, it would be unfair to use the original initializing way. Therefore, we also randomly initialize the OSML model.

**DPMM**: Dirichlet Process Mixture Model (DPMM) (Jerfel et al., 2019) employs a Chinese Restaurant Process to conduct the mixture meta-knowledge with a dynamic number of components. Note that it is not a Bayesian method and employs the point estimation to update the meta-knowledge.

**BOMVI**: Bayesian Online Meta-Learning with Variational Inference (BOMVI) (Yap et al., 2021) is a state-of-the-art algorithm, which conducts a meta-knowledge distribution to address the catastrophic forgetting issue in continual meta-learning. Similarly, it also employs variational inference to update the meta-knowledge.

**VC-BML**: Variation Continual Bayesian Meta-Learning (VC-BML) (Zhang et al., 2021) is another state-of-the-art algorithm, which also employs a truncated Chinese Restaurant Process to conduct the mixture meta-knowledge. Different from DPMM, it uses the Bayesian inference to conduct the mixture distribution of meta-knowledge and places an upper bound on the number of components to reduce the computational consumption.

All the baselines and our proposed SMM-CML follow the experimental setting as described in Sec. F.3.

## F.2    THE DATASETS

**VGG-Flowers** VGG-Flowers(Nilsback & Zisserman, 2008) consists of 102 flower categories. Also, we randomly choose 66 categories for meta-training, 16 categories for validation and the remained 20 categories for meta-test.

**miniImagenet**:miniImagenet(Ravi & Larochelle, 2017) is designed for few-shot learning, which consists of 100 different classes. Similarly, we also split the dataset into three datasets (i.e., 64 classes for meta-training, 16 classes for validation and 20 classes for meta-test) following the existing works.

**CIFAR-FS**:CIFAR-FS(Bertinetto et al., 2018) dataset used in our experiment is adapted from the CIFAR-100 dataset (Krizhevsky et al., 2009) for few-shot learning, which consists of 100 classes. Following the existing works (Yap et al., 2021; Zhang et al., 2021), we also randomly split the datasets, where 64 classes are used for meta-training, 16 classes are used for validation and the remained 20 classes are used for meta-test, respectively.

**Omniglot**:Omniglot(Lake et al., 2011) is a widely-used dataset, which contains 1,623 different handwritten characters from 50 different alphabets. Following the previous works (Yap et al., 2021; Zhang et al., 2021), we randomly split the dataset into three subsets, 1,100 characters for meta-training, 100 characters for validation and the remaining 423 characters for meta-test.

To create the online non-stationary setting, we assume the above datasets are arriving and available sequentially. Moreover, we focus 5-way 5-shot task, which conducts the low-resource environment. For each dataset, we form the streaming tasks via randomly sampling 5 classes with replacement as a

Table 4: The convolution neural network architecture in SMM-CML and baselines.

| Layers | Output Size |
|---|---|
| Input image | $28 \times 28 \times 3$ |
| The first convolution layers | $14 \times 14 \times 64$ |
| The second convolution layers | $7 \times 7 \times 64$ |
| The third convolution layers | $3 \times 3 \times 64$ |
| The forth convolution layers | $1 \times 1 \times 64$ |

Table 5: Some important hyper-parameters used in our experiments.

| Hyper-parameter | VGG-Flowers | miniImagenet | CIFAR-FS | Omniglot |
|---|---|---|---|---|
| The number of outer update step | 2000 | 2000 | 2000 | 2000 |
| The outer learning rate | 0.001 | 0.001 | 0.001 | 0.001 |
| The number of outer update step | 3 | 3 | 3 | 1 |
| The inner learning rate | 0.05 | 0.1 | 0.1 | 0.1 |

task. And we randomly sample 5 examples for each class in a support set and 15 examples for each class in a query set.

In our experiment, we also consider another more challenging setting, where tasks from different datasets are mixed and arrive one by one. In this setting, we conduct different tasks stream with a length of 100, and then train the model on each task one by one and evaluate the performance on each dataset.

### F.3 THE DETAILS OF EXPERIMENT SETTING

For each task, we employ the same convolution network as our base network following the previous works (Yap et al., 2021; Zhang et al., 2021), which is showed in Tab. 4. For our model, we use the Adam optimizer as the outer optimizer and the SGD optimizer as the inner optimizer. For the Monte Carlo sampling used in our algorithm, we set the number of sampling as 5. For the initial number of components in the multi-modal meta-knowledge, we set it as 4. All the important hyper-parameters can be seen in Tab. 5. We ran our algorithm on NVIDIA Tesla V100 32GB GPU. It took about 54 hours to train.

### F.4 ADDITIONAL EXPERIMENTAL RESULT

In what follows, we present the full result on the streaming datasets (i.e., VGG-Flowers, miniImagenet, CIFAR-FS and Omniglot), and change the order of datasets to verify the generality of our algorithm.

#### F.4.1 META-TEST ACCURACIES ON EACH DATASET AT DIFFERENT META-TRAINING STAGE

We only show the average result at each meta-training stage and the performance on each dataset at the last meta-training stage in the main text. We additionally show the full results in Tab. 6. Although SMM-CML can not achieve the best performance on all having arriving datasets at some meta-training stages (i.e., *CIFAR-FS* and *miniImage*), it outperforms all the baselines on the average results, which confirms the effectiveness of SMM-CML. Additionally, SMM-CML can not only maintain the performance on the old datasets, but also achieve better results on the new datasets, which illustrates that it can alleviate better catastrophic forgetting than other baselines. Note that SMM-CML achieves the best performance on the current datasets at each stage (especially compared to VC-BML, where it assumes that each component is mutually exclusive), which shows that our proposed model can resolve the conflict between the learned meta-knowledge and the incremental meta-knowledge, and it is expected that the multi-modal can utilize the shared meta-knowledge to improve the performances.

Tab. 6 also shows the detailed results of SMM-CML before and after evidential sparsification. The results show that SMM-CML still achieves a comparative performance on most datasets at each

Table 6: Performance of our SMM-CML and the baselines on each datasets at each meta-training stage. The best performance on each dataset is marked with boldface and the second best is marked with underline.

| Meta-Training Stage | Algorithms | VGG-Flowers | miniImagenet | CIFAR-FS | Omniglot | Average |
|---|---|---|---|---|---|---|
| VGG-Flowers | FTML | $76.84 \pm 1.75$ | - | - | - | $76.84 \pm 1.75$ |
| | OSML | $79.61 \pm 1.50$ | - | - | - | $79.61 \pm 1.50$ |
| | DPMM | $78.97 \pm 1.52$ | - | - | - | $78.97 \pm 1.52$ |
| | BOMVI | $77.05 \pm 1.80$ | - | - | - | $77.05 \pm 1.80$ |
| | VC-BML | $\underline{83.71 \pm 1.58}$ | - | - | - | $\underline{83.71 \pm 1.58}$ |
| | SMM-CML | $\mathbf{85.11 \pm 1.46}$ | - | - | - | $\mathbf{85.11 \pm 1.46}$ |
| miniImagenet | FTML | $76.51 \pm 1.92$ | $44.97 \pm 1.77$ | - | - | $60.74 \pm 1.85$ |
| | OSML | $76.19 \pm 1.68$ | $56.11 \pm 1.77$ | - | - | $66.15 \pm 1.73$ |
| | DPMM | $\underline{76.65 \pm 1.79}$ | $56.45 \pm 1.74$ | - | - | $66.55 \pm 1.77$ |
| | BOMVI | $75.75 \pm 1.97$ | $45.12 \pm 1.74$ | - | - | $60.44 \pm 1.86$ |
| | VC-BML | $76.47 \pm 1.41$ | $\mathbf{59.71 \pm 1.75}$ | - | - | $\underline{68.09 \pm 1.58}$ |
| | SMM-CML | $\mathbf{81.71 \pm 1.42}$ | $\underline{57.19 \pm 1.66}$ | - | - | $\mathbf{69.45 \pm 1.54}$ |
| CIFAR-FS | FTML | $75.11 \pm 1.84$ | $54.89 \pm 1.66$ | $70.13 \pm 2.07$ | - | $66.71 \pm 1.86$ |
| | OSML | $78.29 \pm 1.63$ | $57.36 \pm 1.54$ | $69.07 \pm 2.01$ | - | $68.24 \pm 1.73$ |
| | DPMM | $75.60 \pm 1.76$ | $55.79 \pm 1.75$ | $70.15 \pm 2.07$ | - | $67.18 \pm 1.86$ |
| | BOMVI | $74.08 \pm 1.60$ | $47.55 \pm 1.84$ | $57.07 \pm 1.86$ | - | $59.57 \pm 1.77$ |
| | VC-BML | $\underline{79.04 \pm 1.54}$ | $\mathbf{59.17 \pm 1.74}$ | $\underline{71.40 \pm 1.93}$ | - | $\underline{69.87 \pm 1.74}$ |
| | SMM-CML | $\mathbf{79.29 \pm 1.48}$ | $\underline{58.98 \pm 1.65}$ | $\mathbf{73.89 \pm 1.69}$ | - | $\mathbf{70.72 \pm 1.61}$ |
| Omniglot | FTML | $63.04 \pm 2.01$ | $37.27 \pm 1.69$ | $47.95 \pm 1.99$ | $99.31 \pm 0.28$ | $61.89 \pm 1.49$ |
| | OSML | $70.68 \pm 1.83$ | $40.67 \pm 1.50$ | $51.89 \pm 2.04$ | $\underline{99.35 \pm 0.24}$ | $65.65 \pm 1.40$ |
| | DPMM | $65.20 \pm 1.67$ | $\underline{48.53 \pm 1.63}$ | $60.15 \pm 2.30$ | $99.16 \pm 0.29$ | $68.26 \pm 1.47$ |
| | BOMVI | $\mathbf{73.19 \pm 1.86}$ | $46.28 \pm 1.62$ | $58.99 \pm 2.14$ | $97.71 \pm 0.53$ | $69.04 \pm 1.54$ |
| | VC-BML | $71.02 \pm 1.76$ | $48.53 \pm 1.82$ | $59.14 \pm 2.01$ | $99.21 \pm 0.47$ | $\underline{69.48 \pm 1.52}$ |
| | SMM-CML | $\underline{71.92 \pm 1.86}$ | $\mathbf{50.07 \pm 1.66}$ | $\mathbf{64.50 \pm 1.83}$ | $\mathbf{99.36 \pm 0.22}$ | $\mathbf{71.46 \pm 1.39}$ |

meta-training stage, compared to before evidential sparsification. It further confirms the effectiveness of our proposed evidential sparsification.

### F.4.2 ADDITIONAL EXPERIMENTAL IN DIFFERENT ORDER

To further confirm the generality of our model, we change the order of datasets in the streaming tasks. We conduct the experiments on a new order, where the model is trained chronologically on *Omniglot*, *CIFAR-FS* , *miniImagenet* and *VGG-Flowers*. The results are shown in Tab. 7. The result on the streaming tasks with a different order shows that SMM-CML still outperforms other baselines, which further confirms the generality of SMM-CML.

Table 7: Performance of our SMM-CML and the baselines on each datasets at each meta-training stage. The best performance (without 'original') on each dataset is marked with boldface and the second best (without 'original') is marked with underline.

| Meta-Training Stage | Algorithms | Omniglot | CIFAR-FS | miniImagenet | VGG-Flowers | Average |
|---|---|---|---|---|---|---|
| Omniglot | FTML | $99.25 \pm 0.24$ | - | - | - | $99.25 \pm 0.24$ |
| | OSML | $98.20 \pm 0.39$ | - | - | - | $98.20 \pm 0.39$ |
| | DPMM | $97.15 \pm 0.48$ | - | - | - | $97.15 \pm 0.48$ |
| | BOMVI | $97.35 \pm 0.73$ | - | - | - | $97.35 \pm 0.73$ |
| | VC-BML | $\underline{99.28 \pm 0.48}$ | - | - | - | $\underline{99.28 \pm 0.48}$ |
| | SMM-CML | $\mathbf{99.43 \pm 0.21}$ | - | - | - | $\mathbf{99.43 \pm 0.21}$ |
| | original | $99.31 \pm 0.25$ | - | - | - | $99.31 \pm 0.25$ |
| CIFAR-FS | FTML | $96.12 \pm 0.76$ | $67.08 \pm 1.87$ | - | - | $81.60 \pm 1.32$ |
| | OSML | $96.09 \pm 0.53$ | $66.20 \pm 2.02$ | - | - | $81.15 \pm 1.28$ |
| | DPMM | $93.31 \pm 0.80$ | $60.88 \pm 2.03$ | - | - | $77.10 \pm 1.42$ |
| | BOMVI | $\underline{97.68 \pm 0.43}$ | $56.29 \pm 2.00$ | - | - | $76.99 \pm 1.22$ |
| | VC-BML | $\mathbf{97.72 \pm 0.38}$ | $\underline{72.8 \pm 1.74}$ | - | - | $\underline{85.26 \pm 1.06}$ |
| | SMM-CML | $97.66 \pm 0.39$ | $\mathbf{75.39 \pm 1.71}$ | - | - | $\mathbf{86.53 \pm 1.05}$ |
| | original | $98.15 \pm 0.39$ | $73.82 \pm 1.75$ | - | - | $85.99 \pm 1.07$ |
| miniImagenet | FTML | $96.63 \pm 0.58$ | $68.60 \pm 1.79$ | $54.68 \pm 1.9$ | - | $73.30 \pm 1.42$ |
| | OSML | $95.04 \pm 0.79$ | $\underline{69.20 \pm 1.72}$ | $55.13 \pm 1.81$ | - | $73.12 \pm 1.44$ |
| | DPMM | $95.01 \pm 0.67$ | $64.93 \pm 2.14$ | $55.49 \pm 1.74$ | - | $71.81 \pm 1.52$ |
| | BOMVI | $\underline{97.01 \pm 0.70}$ | $59.25 \pm 1.76$ | $46.21 \pm 1.66$ | - | $67.49 \pm 1.37$ |
| | VC-BML | $96.29 \pm 0.58$ | $69.05 \pm 1.68$ | $\underline{59.25 \pm 1.86}$ | - | $\underline{74.86 \pm 1.37}$ |
| | SMM-CML | $\mathbf{97.10 \pm 0.46}$ | $\mathbf{70.67 \pm 1.97}$ | $\mathbf{59.97 \pm 1.70}$ | - | $\mathbf{75.91 \pm 1.38}$ |
| | original | $96.74 \pm 0.69$ | $71.35 \pm 1.72$ | $60.13 \pm 1.80$ | - | $76.07 \pm 1.40$ |
| VGG-Flowers | FTML | $93.69 \pm 0.83$ | $58.27 \pm 1.81$ | $45.75 \pm 1.52$ | $80.32 \pm 1.77$ | $69.51 \pm 1.48$ |
| | OSML | $91.79 \pm 1.14$ | $\underline{59.05 \pm 1.80}$ | $46.51 \pm 1.64$ | $81.71 \pm 1.69$ | $69.77 \pm 1.57$ |
| | DPMM | $93.21 \pm 1.03$ | $\mathbf{61.55 \pm 1.82}$ | $45.01 \pm 1.56$ | $80.71 \pm 1.72$ | $70.12 \pm 1.53$ |
| | BOMVI | $\mathbf{97.13 \pm 0.54}$ | $58.77 \pm 1.89$ | $\underline{47.24 \pm 1.81}$ | $75.59 \pm 2.04$ | $69.68 \pm 1.57$ |
| | VC-BML | $92.80 \pm 0.82$ | $58.36 \pm 1.87$ | $47.09 \pm 1.78$ | $\underline{82.92 \pm 1.46}$ | $\underline{70.29 \pm 1.48}$ |
| | SMM-CML | $\underline{94.07 \pm 0.84}$ | $58.76 \pm 1.64$ | $\mathbf{48.74 \pm 1.47}$ | $\mathbf{83.31 \pm 1.50}$ | $\mathbf{71.22 \pm 1.36}$ |
| | original | $93.25 \pm 0.81$ | $58.94 \pm 1.82$ | $49.29 \pm 1.74$ | $84.16 \pm 1.56$ | $71.41 \pm 1.48$ |

