# OpenReview forum: "Scalable Multi-Modal Continual Meta-Learning"
_ICLR.cc/2023/Conference — Submitted to ICLR 2023_

### Official Review · Reviewer_MeHS · 2022-10-27

**Confidence:** 3
**Correctness:** 3
**Technical Novelty And Significance:** 3
**Empirical Novelty And Significance:** 2
**Recommendation:** 5

**Clarity, Quality, Novelty And Reproducibility:**

I think the paper has a lot of room for improvement in terms of presentation. The idea of applying evidential learning to choose meta-knowledge components is interesting. There are some concerns regarding experiments.

**Strength And Weaknesses:**

Strength

-Few shot continual learning is a hot research topic, the authors successfully identify two weaknesses of the existing works, and proposed a principal solution to solve them.

-Evidential theory is a well suited equipment to solve the mutual exclusiveness among mixture components.

-Detailed theoretical derivation is provided.



Weakness

-I have a little bit of hard time following this paper. For instance, the title highlights “scalable” and “multi-modal”. However, I do see much description on these two terms, except in title and abstract. Not sure the meaning of “multimodal”. “Scalable” I guess it refers to the ability of filtering out the components. However, the current experiments do not well support this claim.

Another part is the technical description in evidential theory. I am not clear how to map the concepts in evidential theory to continual meta learning problem without explanation. This prevents me from understanding the technical details intuitively. Also, some terms in evidential theory lacks definition, e.g. focal set in page 5.



-In page 1, the authors use user profiling as an example to explain the unreasonable assumption on mutual exclusiveness of meta knowledge components. However, the chosen experimental datasets don’t reflect this to my understanding. Please explain (or better qualitatively illustrate) how the proposed method addresses the mutual exclusiveness on these test data.



-The experimental results of VC-BML in Table 2 seems lower than the reported numbers in the original paper. Please clarify if there is any difference in experimental setting.



-Regarding the scalable claim, I would recommend conducting longer sequence of tasks/datasets to show the value of the proposed m method. The current reported results show a marginal improvement over existing methods.

**Summary Of The Paper:**

This paper studies the continual meta learning, widely used in low-resource setting.  Compared with the existing works, this work is motivated by two points. One is existing works assume the number of components of meta knowledge is mutually exclusive. Two, existing works usually only use a prior determined by Chinese Restaurant Processs, but do not make a posterior decision on number of components. To address the two issues, this method proposes an IBP prior to determine whether to increase the number of components, then leverages evidential theory to filter out the uninformative components.


--- post-rebuttal ---\
I have read the authors' response and other reviewers' comments. I think the current form of this paper is not ready for ICLR publication.

**Summary Of The Review:**

Overall I think the idea of applying evidential theory to address the inappropriate mutual exclusiveness of mixture meta knowledge components is interesting. While unfortunately there are several concerns on the presentation and experiments, that needs to be addressed and clarified.

---

> ### Author Response · Authors · 2022-11-18
> **Response to reviewer MeHS**
>
> Thanks for the positive comments and the detailed suggestions. We hereby respond to the comments as follows:
>  1. "Multi-modal" and "scalable" highly summarize our two main contributions. "Multi-modal" refers to the meta-knowledge for each cluster of similar tasks, which follows a statistical distribution of values with multiple peaks [1]. "scabable" means that our proposed algorithm employs the evidential sparsity method to determine the posterior of meta-knowledge components via removing the redundant ones. Figure 3 illustrates the underlying relationship between components and tasks in our model, and Section 5.3 explores the effectiveness of our scalable meta-knowledge. We add the explanation of these two terms in the forth paragraph of the Introduction Section of the revised up-to-date paper.
> 2. It is a good suggestion to give some intuitive explanations to help readers understand our evidential sparsity method better. We add one more figure, i.e., Figure 6, in  Appendix C to give some intuitive understanding, and explain briefly here to make the reviewer understand better:
>
>     The key idea of evidential theory is to merge evidences from different sources. In our evidential sparsity, we cast beta distributions at each time as independent evidences. For each timestep, a beta distribution represents the relationship between the tasks and the certain meta-knowledge component. The application of evidential theory helps to fuse each piece of evidences accociated with each component at both the current time and the previous time. Specifically, each independent evidence provides support or doubt information for the related meta-knowledge component, and also provides the measure of uncertainty (i.e., the probability of the universal set {Z}). Therefore, the fused results contain the information of each independent beta distribution from previous time and the current time, so that it can filter out the redundant components which are neither related to the current tasks nor the previous tasks.
>
>     The above explanation and the typos across the whole paper have been included and corrected in the revised up-to-date draft.
> 3. Many transfer learning works [2] have illustrated the effectiveness of shared knowledge in image classification, while there are some differences between the grayscale (i.e., Omniglot) and color image, or fine-grained (i.e.,  VGG-Flowers) and coarse-grained classification (i.e., miniImagenet and CIFAR-FS). Our model allows tasks to be associated with a subset of meta-knowledge components instead of a single meta-knowledge component, which enables both the sharing (i.e., the intersection) and the diversity (i.e., the remaining components) of meta-knowledge. It has been also illustrated by the Figure 3.
> 4. The main differences on the hyperparameter setting of the baseline between our paper and the original paper are the outer learning rate and the weight of KL term. To ensure the generalization, we set the same value for the outer learning rate and the weight of KL-term across different training stage, respectively. It improves the model generalization in the dataset stream, since the specific setting of hyperparameters for one certain dataset in the datastream weakens the generalization of the algorithm.
> 5. We have conducted an experiment where models are trained on a long task sequence, with the length up to 1,000. The task sequence consists of tasks sampled randomly from different datasets. The results in Table 2 have confirmed the effectiveness of our proposed algorithm under the long sequence setting.
>
>     Few-shot continual learning is a practically meaningful topic, and is more challenging than either few-shot learning or continual learning. The average improvement (i.e., 2%) on each dataset can be benefitial to the community, and we hope that the proper integration of IBP and evidential theory could inspire more interesting future works in this filed.
>
> [1] Multimodal model-agnostic meta-learning via task-aware modulation. NeurIPS  2019.
>
> [2] A study on cnn transfer learning for image classification. 2018

---

### Official Review · Reviewer_XxCX · 2022-10-28

**Confidence:** 3
**Correctness:** 2
**Technical Novelty And Significance:** 2
**Empirical Novelty And Significance:** 2
**Recommendation:** 3

**Clarity, Quality, Novelty And Reproducibility:**

Poorly written. The idea is innovative. The reproducibility is questionable given the lack of clarity and no code is provided. The authors did not mention whether they will ever provide code when the paper is published.

**Strength And Weaknesses:**

Strength
+ The idea is innovative and may be of value
+ Based on the reported empirical results, the proposed method seems working.

Weakness
- The paper is poorly written, very difficult to follow. Especially in the mathematic derivation, the overloading and inconsistency choice of notations cause a lot of confusion. For example, in 3.2, what exact A is? Is it an element in Z or a set? The definition of the plausibility of A in Eq. (5) is confusing. In the paragraph following Eq. (7), what does exact k represent, a cluster or component? Awkward sentences and poor choices of worlding show up in so many places.
- If I understand correctly, the posterior calculated to determine whether to remove components is only conditioned on the data at current time point as one has no access to previous data in a continual learning setting. If this is the case, the removal of components could easily lead to catastrophic forgetting.
- There is lack of clarity in the description of the setting of their experiment. Specially, at what point, the performance on a task was evaluated? Was it right after the training for that task was completed or after the training for all tasks were done. If the former, catastrophic forgetting was not properly evaluated.


**Summary Of The Paper:**

This paper presents a scalable multi-model continual meta-learning algorithm. This method associates a cluster of similar tasks with a set of meta-knowledge components instead of one single component in previous approaches. If I understand correctly, “multi-modal” in the title is due to a set of components being used. The authors proposed to use Indian buffet process to determine whether to add new components into the mixture when learning for a new task and leverage evidential theory to remove redundant components from the mixture. They evaluated their methods using four benchmark datasets.

**Summary Of The Review:**

The proposed idea may be interesting. However, poor presentation really affects a thorough understanding of the paper.

---

> ### Author Response · Authors · 2022-11-18
> **Response to reviewer XxCX**
>
> Thanks for the detailed comments. The followings are our repsonses to the comments:
> 1. We have revised the typos of the mathematical derivation to make the paper clearer in the updated version. And we also answer the questions about the notations:
>     - The symbol $A$ refers to a single strict subset, which we have claimed in the paragraph following equation 4 in the original draft.
>     - The symbol $k$ represents the k-th component, and we revise such typos in the updated draft.
>
>     We hope that our revision and explanation could help the reviewer to more easily follow the mathematical derivation.
> 2. Our proposed evidential sparsity method casts each beta distribution of both the current time and the previous time as a piece of evidence (which we have claimed in  Section 4.3 in the original draft). The evidence provided by the beta distribution at the previous time maintains the relationships between the previous data and the meta-knowledge components. Hence, the information from both the previous and current time, helps to avoid the catastrophic forgetting issue, which is also confirmed by the results in Table 3. We revised our draft in Section 4.3 to make this point more clear.
> 3. In the experiment, the performance is evaluated after each meta-training stage, where models are trained on the current dataset or the current task. We choose the average performance at each stage as a main evaluation, since the recent continual learning researches [1] focus on not only the catastrophic forgetting issue, but also the ability to capture the incremental knowledge from newer tasks (which has been claimed in the first paragraph of the Introduction Section). The used average performance reveals the trade-off of these two focuses, and the experimental results shown in table 1 and 2, illustrate the effectiveness of our algorithm in the setting of continual learning. Besides, we also show all results in Table 6 in the Appendix due to the limited space of the main paper.
> 4. We will release the github link of our code to the paper upon the acceptance of our paper.
>
> [1] A continual learning survey: Defying forgetting in classification tasks. PAMI 2021.

---

### Official Review · Reviewer_yxSH · 2022-11-03

**Confidence:** 2
**Correctness:** 3
**Technical Novelty And Significance:** 2
**Empirical Novelty And Significance:** 2
**Recommendation:** 5

**Clarity, Quality, Novelty And Reproducibility:**

The clarity of the paper can be improved. The IBP part looks novel to me but I am not sure about the novelty of the evidential sparsity part.

**Strength And Weaknesses:**

Strength

1. The paper aims at a very interesting and important topic.
2. The use of IBP prior to handle the update of the number of mixture components looks novel.

Weaknesses

1. The writing can be improved. For example, it would help if the authors can elaborate on why the proposed method can allow the sharing of meta-knowledge and why previous methods cannot (as this is one major contribution summarized by the authors). The term "multi-modal" throughout the paper also seems obscure to me. Plus, by "multi-modal knowledge" I guess the authors do not want to mean something like knowledge from image and text but want to mean something relating to informativeness as in [1]. It would help if the authors can make this point more clear.
2. I am not sure about the novelty of section 4.3. In section 4.3, many equations (Eqs 14-17) are very similar to section 2.2 of [1] without explicitly citing [1]. However, I am not an expert in this field. I would like to bring it to other reviewers' attention and would defer to their comments.


[1] Masha Itkina, Boris Ivanovic, Ransalu Senanayake, Mykel J. Kochenderfer, and Marco Pavone. Evidential sparsification of multimodal latent spaces in conditional variational autoencoders. In NeurIPS 2020.

**Summary Of The Paper:**

This paper proposes a continual meta-learning framework, Scalable Multi-Modal Continual Meta-Learning. Specifically, the authors employ the Indian Buffet Process for sharing meta-knowledge across different tasks and encourage evidential sparsity for parameter efficiency. The experiments validate the proposed method under the online non-stationary setting.

**Summary Of The Review:**

Given the strength and weaknesses, I tend to rate this paper as marginally below the acceptance.

---

> ### Author Response · Authors · 2022-11-18
> **Response to reviewer yxSH**
>
> We thank the reviewer for the positive comments and the helpful suggestions. The following is our response to the reviewer's comments:
> 1. In the fourth paragraph of Introduction and Section 4.1, we have discussed why and how our method enables the sharing of meta-knowledge. To make it clearer, we revised the fourth paragraph of the Introduction in the updated draft and gave a brief explanation as follows:
>
>    Our method holds an assumption that the mapping between meta-knowledge components and task clusters depends on k independent Bernoulli distributions, generated from Beta distributions (where K denotes the number of components). So that our method allows a cluster of similar tasks to be associated with a subset of mixture components and thus shares meta-knowledge via the overlap components among different task clusters. In contrast, the previous works assume a categorical distribution, where components are mutually exclusive and each cluster of tasks is associated strictly with a single meta-knowledge component. That assumption prevents sharing meta-knowledge components.
>
> 2. In our work, "multi-modal" means a statistical distribution of values with multiple peaks. This is also adopted by previous papers such as [1] and the wikepedia [3]. Recently, "multi-modal" is frequently used to descibe cases where the model input or output has different data modes such as texts and images. To avoid confusion, we add the above explanation in the updated draft.
>
> 3. Although our equations (Eqs 14-17) in Section 4.3 can be roughly similar to those in Section 2.2 of the mentioned work [2], they are merely the intermediate derivation steps of the sparse meta-knowledge distribution. It is the Equation 19 (i.e., Equation 15 in the updated draft) is one of our core contributions. Also, what we present is to make the mutually independent meta-knowledge components (i.e., the meta-knowledge with K independent Bernoulli distributions) sparse, while [2] focuses on the mutual exclusiveness (i.e., the discrete latent space of CVAE), which is conflit with ours. Moreover, our work provides a novel way to apply evidential theory on the continual setting, and we hope that it would be benefitial to the research community.
>
> [1] Multimodal model-agnostic meta-learning via task-aware modulation. NeurIPS  2019.
>
> [2] Evidential sparsification of multimodal latent spaces in conditional variational autoencoders. NeurIPS 2020.
>
> [3] https://en.wikipedia.org/wiki/Multimodal

---

### Author Response · Authors · 2022-11-18
**General Response and Summary of Updates of Manuscript**

Thank you very much for all reviewers' positive comments and insightful suggestions. The reviewers pointed out the unclear presentation of some concepts (e.g., "multi-modal") as well as typos in the mathematic derivation. We have checked the whole paper, corrected the typos, and updated the draft to make the paper clearer with the following revision:
1. We added more explantation to two concepts (i.e., 'multi-modal' and "scalable") in the fourth paragraph of the Introduction Section and Section 4.1.
    - The term "multi-modal" is used to describe a statistical distribution of values with multiple peaks or components. This is also adopted and widely used by previous papers such as [1] and the wikepedia page [3]. In this paper, we use "multi-modal" to describe the meta-knowledge distribution associted with each cluster of similar task, which has multiple components instead of a single one.
    - The term "scalable" means that although the number of meta-knowledge components increases with the newer tasks encoutering, the redundant components can be removed based on the evidential theory to avoid the umlimited increase issue. In other words, the IBP provides a prior on the number of components and the evidential sparsity makes a posterior decision on the final number of components.

2. We added the discussion of sparsification methods in Section 2 to confirm our contributions of the proposed evidential sparsity. One of our contributions is to sparse the meta-knowledge components according to Equation 15 in the revised draft. What we present is to sparse the mutually independent meta-knowledge components (i.e., the meta-knowledge with K independent Bernoulli distributions), while [1] focuses on the mutual exclusiveness (i.e., the discrete latent space of CVAE). In addition, our work provides a novel application of evidential theory on the continual learning, and we hope that it can provide fresh insights for the research community.

3. We added the intuitive explanation on the application of evidential theory on the continual meta-learning in the second paragraph of Section 4.3 and Figure 6 in Appendix C.

4. We revised typos across the whole draft and minor errors in the mathematical derivation and Figure 2.

We hope the revised up-to-date draft can help reviewers and readers to better understand our work and help to clarify our contributions.

[1] Multimodal model-agnostic meta-learning via task-aware modulation. NeurIPS  2019.

[2] Evidential sparsification of multimodal latent spaces in conditional variational autoencoders. NeurIPS 2020.

[3] https://en.wikipedia.org/wiki/Multimodal

---

### Decision · Program_Chairs · 2023-01-20

**Decision:**

Reject

**Justification For Why Not Higher Score:**

N/A

**Justification For Why Not Lower Score:**

N/A

**Metareview: Summary, Strengths And Weaknesses:**

This paper proposes SMM-CML, a Scalable Multi-Modal MetaLearning algorithm where a cluster of similar tasks are associated with multiple components, allowing tasks to share meta-knowledge while maintaining their diversity. In general, the reviewers agreed that the idea of the paper is interesting. However, the reviewers had many concerns about the technical details, clarification, and experiments used in the paper. The authors addressed some of them but the reviewers are not fully convinced.